

# Phosphorus transport in lateral subsurface flow at forested hillslopes

Jakob Sohrt[1] , Heike Puhlmann[2] and Markus Weiler[1]

[1]Chair of Hydrology, Albert-Ludwigs-University of Freiburg, Friedrichstraße 39, 79098 Freiburg, Germany

[2]Forest Research Institute of Baden-Württemberg, Wonnhaldestr. 4, 79100 Freiburg, Germany

*Correspondence to*: Jakob Sohrt, jakob.sohrt@posteo.de

**Abstract.** This study is concerned with the transport of Phosphorus (P) with lateral subsurface flow in the organic layer and topsoil of three forested headwater sites in Germany. Sampling frequency was set proportional to the incident flow rate in high temporal resolution. With this approach we want to investigate intra-event dynamics of P transport in lateral subsurface flow to establish initial process understanding about this potentially relevant pathway of P loss in forested hillslopes. With the organic layer being an important transfer site in the P cycle of temperate forests, availability and

transportability of P in short timescales may reveal details about the overall balance of P in theses ecosystems. Our results demonstrate that P concentrations in lateral flow are highly variable within and in between distinct flow events as well as among our study sites. To determine possible controls of the P transport we constructed multiple linear models of the P concentration in lateral flow as a function of site specific environmental datasets. Site affiliation was responsible for more than half of total explained variability regarding P concentration in lateral flow, followed by flow

rate, electric conductivity of subsurface lateral flow

Keywords: *phosphorus, transport controls, lateral flow, organic layer, temperate forest, event dynamics*



## Introduction

Phosphorus is essential for all life and its availability often limits primary productivity of ecosystems (Elser et al. 2007; Elser et al. 2000). Original P input into unfertilized, terrestrial ecosystems is mainly caused by weathering of bedrock material and likely to a lesser degree by atmospheric deposition (Buendia et al. 2010; Mahowald et al. 2008; Newman 1995; Tipping et al. 2014) Both fluxes of P into the ecosystem are generally orders of magnitudes smaller than P fluxes within the ecosystem, like plant uptake and litter fall (Cole and Rapp 1981; Gosz et al. 1972; Yanai 1992). This

necessitates the existence of efficient mechanisms to reduce losses of P from the ecosystem (Hättenschwiler and Vitousek 2000; Lang et al. 2016; Wood et al. 1984), which are thought to occur mostly through solutes and suspended particles in soil water seepage, groundwater and eventually stream flow (Cole and Rapp 1981).

The organic layer plays a key role in the P cycle of temperate forest ecosystems: It receives large P inputs through dead biomass from above and below ground sources as well as from atmospheric deposition, which are then available for

biotic turnover. P can be removed from the organic layer again by biological uptake or leaching with vertical or lateral water flow (Attiwill and Adams 1993; Clark et al. 2001; Cole and Rapp 1981; Eckstein et al. 1999; Fisk and Fahey 2001; Gaudinski et al. 2000; O'neill and DeAngelis 1981; Yanai 1992).

In situations where P availability may limit ecosystem productivity, organisms are expected to compete for available P and thus keep its concentrations in soil solution, the point of microbial- and plant uptake, at a low level (Björkman and

Lundeberg 1971; Goldberg 1990; Turner et al. 2013). Additionally, ionic P is effectively immobilized on the surfaces of biological and mineral particles, further reducing the potential of P loss through soil water flow. Particulate forms of P may be retained in the soil through mechanical filtering, but their relative depletion with soil depth has been demonstrated to be lower than for ionic P (Compton and Cole 1998; Fitzhugh et al. 2001; Ilg et al. 2009; Johnson et al. 2016; Kaiser et al. 2001a,b; Qualls et al. 2002). Through these processes, the concentration of P in groundwater and

streamflow of undisturbed forested headwaters is often orders of magnitudes lower than in streamflow generated at fertilized sites, illustrating the tightness of forested headwaters P cycle (Beaulac and Reckhow 1982; Dillon and Kirchner 1975; Reckhow et al. 1980; Vuorenmaa et al. 2002).

Lateral subsurface flow is an important process of runoff generation in forest soils, due to the regular occurrence of preferential flowpaths and the uneven distribution of precipitation input to the organic layer (Bachmair et al. 2012;

Schwärzel et al. 2012). Preferential flowpaths may develop along roots and animal burrows which are a common feature of forest soils, or through soil physical processes like freezing, thawing and changes in soil water content (Aubertin 1971; Beven and Germann 1982; Beven and Germann 2013). The forest canopy is effective in partitioning incoming precipitation, creating areas of higher water input directly at the tree base through stemflow, and at arbitrary positions where throughfall may be accumulated through temporal interception and channeling along canopy elements

(Keim, Skaugset & Weiler 2005; Schwärzel et al. 2012). Other than P transport with water flow through the soil matrix, lateral subsurface flow can effectively transport particulate matter (Bachmair et al. 2009; Cox et al. 2000; Fuchs et al. 2009; Jardine et al. 1990; Vinther et al. 1999). This is highly relevant for the transport of P, as P in the soil water is predominantly found in particulate forms (Timmons, Verry, Burwell & Holt 1977; Cox et al. 2000; Kaiser et al. 2001b; Qualls et al. 2002; Kaiser & Guggenberger 2003; Fuchs et al. 2009; Bol et al. 2016; Julich, Julich & Feger 2016;

Missong et al. 2016). Additionally, the higher flow velocity and reduced contact time between soil material and water in preferential flowpaths reduces retention of transported P on soil particles (Jarvis 2007). The biogeochemical attributes of preferential flowpaths may differ extensively from those of the bulk soil (Aubertin 1971). Earthworm burrows are coated with a substance that is much less effective in retaining inorganic P forms than the bulk soil, and flowpaths along roots – living or dead – are hot-spots of microbial activity (Bundt et al. 2001; Jensen, Hansen & Magid 2002). In





comparison to other forest biomass compartments, microbes – especially bacteria - have a high P content and high turnover rate. During low flow velocities, they may take up P from subsurface flow, or getting dragged along at higher flow velocities (Vinther et al. 1999).

The aim of the study is to improve our understanding of the processes relevant to P transport via lateral flow in the organic layer, which is so far very limited. It is still unclear, to which degree P transport in lateral subsurface flow in the
upper soil layer and organic layer is controlled by availability of transportable P or by the transport capacity transport in the form of lateral subsurface flow. To investigate the importance of these two fundamental processes, we sample P concentration in lateral subsurface flow as well as the flow rate and soil moisture, which may serve as proxies for the transport capacity and small scale hydrological connectivity. We assume, that availability and possible buildup of transportable P is controlled by biological activity resulting in the breakdown of organic layer material, for which air
temperature, soil moisture and the period between rainfall events are used as proxies. To test, whether findings are significant beyond the scope a single site, the experiment is repeated at three sites.

## Methods

Sampling took place at three forested hillslope sites in Germany: Mitterfels (MIT) in Bavaria, Vessertal (VES) in
Thuringia and Conventwald (CON) in Baden-Württemberg. The sites were selected for similar vegetation but differences in mineral soil P content within the framework of the SPP 1685 program on forest phosphorus nutrition (Lang et al. 2016).
All three sites are old-growth european beech (*Fagus sylvatica*) forests with little management in the last decades and sparse or absent undergrowth aside from beech saplings. Sample collection was carried out from March to November
2015 with the construction of the field installations taking place the year before. Recent studies are available that broach issues of P cycling at the same sites as our study with regards to microbes (Bergkemper et al. 2016a,b; Bünemann, Augstburger & Frossard 2016; Zederer, Talkner, Spohn & Joergensen 2017), soil P species (Missong et al. 2016; Prietzel, Klysubun & Werner 2016; Lang et al. 2017), hydrology (Julich et al. 2016; Julich et al. 2017), plants (Netzer, Schmid, Herschbach & Rennenberg 2017) and the effect of liming and nitrogen input (Holzmann et al. 2016).

-- Table 1 --

The capacity of lateral subsurface flow in forest soils to transport large amounts of P in short timescales has been demonstrated by field studies, though the reasons for high spatio-temporal variations in lateral flow P concentrations are
so far not explicitly clear. (Cox et al. 2000; Fuchs et al. 2009; Timmons et al. 1977). Lateral subsurface flow events show a high temporal variability in terms of flow rate (Bachmair and Weiler 2012a; Bachmair et al. 2009) and chemical composition (Burns et al. 1998; Hill et al. 1999; Newman et al. 1998), and may thus not be accurately represented by grab sampling, bulk sampling or suction-based sampling of soil water. To overcome these limitations, we sample free draining, unfiltered lateral flow in the organic layer with a temporal resolution from minutes to hours, to match
temporal dynamics of other parameters related to lateral flow such as precipitation input and changes in soil moisture.

At each hill slope site, a 10 m wide trench was dug along the local contour line. Lateral flow from the organic layer, which varied in thickness from 10 to 25 cm, is captured in this trench and channeled to a sampling point. This is archived by placing a drainage mat to the up-slope wall of the trench, which is connected to a pipe at the bottom. After the installation of the drainage equipment, the trench was filled in again. A more detailed description of the building





process can be found in Bachmair and Weiler (2012b). This method allows for the collection of representative and undisturbed water samples, as no significant filtering occurs. When lateral flow is captured in the trench, the water is directed to a tipping bucket with 100 ml resolution and a 2 L reservoir below. The content of the reservoir is pumped to a flow through cell with a volume of 300 ml when a pre-defined volume of lateral flow water has accumulated. In the flow through cell, electric conductivity (EC) is measured with a *Decagon* CTD-sensor, after which a water sample is

drawn to a 100 ml glass bottle in an enclosed autosampler. 40 samples can be drawn by one autosampler at minimum intervals of 10 minutes. The process is fully automated and all data is transmitted in near real time via the cellular network.

At the field site CON, a meteorological station is available in a distance of less then 100 m, at the site MIT the distance to the nearest meteorological station is approximately 500 m. At the site VES there was no meteorological station in the

direct vicinity and the nearest DWD station 10 km away was used. These meteorological stations are used as sources for time series of air temperature and precipitation. Soil moisture in the organic layer is measured in 3 locations, 2 m up slope of the trench installations.

The water samples were collected in the field twice a week at the site CON and every three weeks at the sites VES and MIT. Collection intervals of this length must be assumed to be too long to allow for the subsequent analysis of different

P species, as biological transformation in untreated samples will change their distribution in much shorter timescales. As a consequence, only total phosphorus of the unfiltered samples ($P_{tot}$) was determined.

While we did attempt to provide full coverage over the sampling period, equipment breakdowns or sample contamination – mostly by insects which drowned in the sample bottles – were a frequent occurrence. We successfully collected samples for about 40 % of the totally occurring flow events.

Samples are digested according to the *US EPA ESS digestion method 310.2 (US EPA 1992)* with the addition of sulfuric acid ($H_2SO_4$), ammoniumperoxidisulfate (($NH_4$)$_2S_2O_8$) and pressurized heating to 130 °C in an autoclave. To solubilize P from possible biological growth on the surfaces of sampling bottles, part of the sulfuric acid used for the digestion was added directly into the sampling bottle, acidifying the sample to pH 0.7. The acidified samples are then shelved for three weeks and placed in to a mechanical shaker once every week, 12 hours at a time, after which the remaining steps

of the digestion process are carried out.

After the digestion, sample pH is adjusted to pH 7 with ultra pure NaOH solution. This is done, because the pH of samples varied between 0.7 and 1.2 after the digestion, which can influence the analysis method and also cause interferences with Si and As.

$P_{tot}$ is measured with the molybdenum blue method by Murphy and Riley (1962) adapted by Drummond and Maher

(1995) using a Autolab 4 device by Green Eyes LLC. Every sample is measured 3 times, standards are interspersed at a rate of 3 standards every 14 samples. Deviations among repeated measurements on the same sample were usually below 5 % of the measured value. From the repeated measurements, the range of the measurement error is calculated as the standard deviation. The error range for P loads in flow events is calculated as the accumulated total measurement error.

To avoid outside contamination during sampling and analysis as much as possible, only dedicated equipment, sample

containers and lab space was used, which were cleaned with deionized water and phosphate free detergent before each use. When the possibility of outside contamination was apparent, the respective samples were discarded. Possible systematic contamination was monitored through the inclusion of standards in every aspect of sample handling. This includes standards being placed in randomly chosen, open sample bottles in the autosamplers at the field sites for the time spans between sample collection dates. The standards are also subjected to the same treatment as the actual

samples with regard to acidification, storage and digestion. From the point of collection to the final measurement of $P_{tot}$, a mass balance is documented for each sample to correct for evaporation losses during the digestion and dilution



through added reagents.

For each hillslope site, time series of lateral flow rate, electric conductivity (EC), air temperature and soil moisture in the organic layer were compiled for the observation period. Intra-event data and event-wise aggregated data (called inter-event data from here on) is used to construct multiple linear models with the $P_{tot}$ concentration in lateral flow as the dependent variable. Depending on the parameter, aggregated inter-event data consists of sums (flow volume), arithmetic averages (flow rate, soil moisture, air temperature) or flow weighted averages (EC, $P_{tot}$) of the original data. Flow events are defined according to two criteria; If only a single peak in lateral flow rate occurs within a distinct precipitation event and until the cessation of lateral flow, the whole period is regarded as a single event. If multiple flow peaks occur within a period of uninterrupted flow, the period may be split into multiple events. If the flow rate falls below 10 % of the preceding flow peak before increasing again and if this increase corresponds to a prior increase in rainfall intensity, this is used as a cutoff point between events.

To identify environmental variables that may explain the observed intra- and inter-event variability in P concentration in the lateral subsurface flow and to test, whether similar relations can be established for all three sampling sites, multiple linear models are utilized. By comparing intra- with inter even data, we want to test to which extend our high resolution sampling is helpful to improve process understanding of lateral P transport.

The parameter selection for inter- and intra-event models respectively was carried out by stepwise forward selection/backwards elimination of variables with respect to the *Akaike Information Criterion* (AIC) through the step()-function in the R base-environment (R-Core-Team 2014). The full set of parameters from which the algorithm choses a reduced set, contains both "real time" indicators, such as lateral flow rate, EC and soil moisture, as well as "between-event" indicators like time duration between events, such as average soil moisture and temperature since the last event (see Tab. 3). The IDs of the three experimental sites are included as a categorical variable, which can also serve as an error term. The "between-event" parameters are exactly the same for both model approaches, while the "real-time" parameters are represented by average values in the inter-event model and the respective raw data in intra-event models. The relative parameter importance was calculated according to (Grömping 2006) in the R-environment.

# Results

Our observations cover the full vegetation period at the the sites MIT and CON and start shortly after leafs shoot at the site VES (Tab. 2). Most precipitation events > 1 mm produced measurable lateral flow, totaling 151 discrete subsurface flow events during the observation period on all 3 sites. Mostly due to field equipment breakdowns, only 62 of these events with a total of 417 individual samples were successfully analyzed for $P_{tot}$.

While the amount of total precipitation during the observation periods was relatively similar among the three sites, the amounts of lateral flow were very different, with 7 times higher flow volume at the site MIT in comparison to site CON and the site VES almost 3 times higher in comparison to CON (Tab. 2). Since our design does not allow for the determination of the source area of lateral flow, it can only be assessed as a total amount, and not as a fraction of the respective rainfall event.

-- Table 2 --

Soil moisture responds to precipitation events in most instances, but at the sites MIT and VES there is a regular occurrence of precipitation events that do not seem to cause changes volumetric water content in the organic layer and topsoil. Overall, there is a marked decline in soil moisture in the organic layer from mid-June to the end of the





vegetation period at the sites MIT and CON (Fig. 1). The $P_{tot}$ concentration in the lateral flow samples ranges from 5 mg l-1 to below the detection limit of our method at 10 µg l-1. Highest $P_{tot}$ concentrations were predominantly found at

the site CON, the lowest at the site VES. The range of $P_{tot}$ concentrations in samples from a single event can be as large as the range of average event concentrations throughout the measurement period at a given site. The existing range of occurring $P_{tot}$ concentrations allowed for the estimation of P loads during events with only a few percent of uncertainty in most cases (Fig. 2).

The average event concentration of $P_{tot}$ in lateral flow does not show a pronounced seasonal pattern and can differ

considerably among successive events (Fig. 1).

The variation of intra-event $P_{tot}$ concentrations appears more ordered, with recognizable periods of increase or decrease (Fig. 2). Still, the underlying processes are not immediately recognizable, as neither flow rate, soil moisture or EC appear to be in a consistent relation with the Ptot concentration. For some events, the $P_{tot}$ concentration decreases consistently over time, irregardless of changes in the flow rate, while being both positively and negatively correlated to

EC (Fig 2 B,C,D). In other events, multiple peaks in the Ptot concentration are evident without a clear relation to flow peaks in the flow rate (Fig 2 A,E).

-- Figure 1 –

-- Figure 2 --

-- Figure 3 –

-- Table 3 --

Results of the multiple linear models, fitted to both intra- and inter-event Ptot concentrations of lateral flow are displayed in Fig. 3 and Tab. 3. In both cases, differences between the study sites are responsible for most of the variability in $P_{tot}$ concentrations of lateral flow explained by the model. The final models, were parameters were selected

according to the respective model AIC, contain different parameters for the intra- and inter-event data models. The total explained variability is notably higher for the model fitted to inter-event data.

## Discussion

Across our sites, most measurable precipitation events produced lateral flow in turn, even when no changes in soil moisture were recorded. This is a strong indication that lateral flow is an important characteristic of the hydrological

system in the organic layer and topsoil of the experimental sites. Still, there are notable differences among the sites regarding the generation of lateral flow following rainfall events. At the site MIT there is a regular occurrence of precipitation events in the summer months that do not coincident with an increase in soil moisture, but which do cause lateral flow. At the sites VES and CON, soil moisture is increased following rainfall events nearly without exception, while lateral flow is not always generated, especially in the drier summer months at the site CON. This indicates, that

the fractions of rainfall that contribute to lateral and vertical flow respectively may differ between the study sites depending on the hydrological setting. The observation that generation of lateral flow is more pronounced at the site MIT than at the other sites is supported further by the observation, that the total amount of lateral flow relative to rainfall is by far highest at this site. The relatively high yields of lateral flow at the site MIT despite the fact that this sites has by far the lowest slope angle (see Tab. 1) could be an indication of hydrophobicity in the organic layer and

topsoil, which might prevent the soil material from retaining water, which in turn would increase potential for lateral subsurface flow. The possible occurrence of hydrophobicity in the dry period at the site MIT might also explain that average P concentrations in lateral flow at this site are significantly lower in the dry period compared to the wet period before: Lower contact times of infiltrating water with soil material as well as a smaller contact area and less exchange



with pre-event soil water would be expected in during periods with hydrophobic soil material, all of which may reduce
the potential for P mobilization (Bogner et al. 2008; Dinh 2017).

The very high temporal variability of the flow rate, with peaks often lasting for some minutes only, confirms the need for high temporal resolution of measurements to adequately observe this parameter. Depending on the research in question, precipitation should also be measured in a comparative resolution and directly on site. Our data shows, that 30 minute resolution in precipitation measurements is not enough to match the temporal variability of lateral flow
generation. At the site MIT, a distance of less than two kilometers between our field site and the meteorological station apparently was high enough to cause situations where lateral flow was measured up to an hour before precipitation onset at the rain gauge. At he field site CON, where the distance between our field site and the meteorological station was only about 100 m and the temporal resolution of the precipitation measurements was 5 minutes, deviations between the onset of lateral flow and precipitation were very low. These differences in the quality of precipitation data among
out sites prohibited us from using precipitation as a model parameter.

Our data clearly demonstrates the existence of high temporal variability in $P_{tot}$ concentrations over the course of single events as well as in average concentrations among distinct flow events. This shows that bulk sampling of single lateral flow events does not allow for generalization about average event conditions. Intra event changes in subsurface flow $P_{tot}$ concentration do seem to represent orderly patterns with distinct phases of $P_{tot}$ concentration in- or decreases. While
outliers, where the $P_{tot}$ concentration of single samples highly deviates from the rest of the event do exist, a sampling frequency from 10 to 20 minutes appears to be high enough to capture the overall event $P_{tot}$ dynamic.

Overall, lateral flow sampling in high temporal resolution actually decreases the ability of our model to predict $P_{tot}$ concentrations in lateral flow when compared to the model fitted to aggregated data, which is meant to represent the
results of a fictional bulk sampling approach. The high significance of both models and the minor difference between R² and adj. R² in the respective models (see Tab. 2) make it appear unlikely, that the additional explained variability in the inter-event data model is simply due to a reduction in sample size, or over-parametrization. Including terms for interactions between the model co-variables soil moisture, temperature and time span in between events did not significantly improve either model.

The AIC-dependent parameter selection algorithm chose somewhat different parameters for intra- and inter-event models. Both model structures are alike in such a way, that differences between sites (included in the model as a categorical variable) are responsible for more than half of the total explained variability. This implies that under given conditions, consistent differences in lateral subsurface flow P concentrations exist among our experimental sites.

For both models, antecedent soil moisture correlates significantly with $P_{tot}$ concentration in lateral flow. Since the
estimate of this parameter is positive in both cases, the models suggests that higher antecedent soil moisture is associated with higher $P_{tot}$ concentration in lateral flow under given conditions. While the cumulative within-event flow volume is an important parameter in the intra-event model, total event volume is not significantly correlated to average event concentration of $P_{tot}$ in the inter-event model. This indicates, that while high-volume events are not significantly associated with overall lower Ptot concentrations, there is a significant decrease of the $P_{tot}$ concentration during the
progression of individual events, leading on average to higher concentrations at the event start in comparison to the end. Another notable discrepancy between the two models is the effect of EC, with accounts for 18 % explained variability in the inter-event model, but only for 4 % in the intra-event model. This would imply, that the average values of EC during events are correlated to a higher degree to average flow and $P_{tot}$ concentration, than their respective variation on intra-event timescales.

The negative correlation of the maximum flow rate during a respective event with $P_{tot}$ in the inter-event model, as well



as the low significance and explained variability of the actual flow rate in the intra-event model suggest, that immediate mobilization of transportable P forms in the organic layer may not be a critical process in lateral flow $P_{tot}$ transport and that higher flow rates may rather cause dilution of P in lateral flow than lead to increased mobilization.

The fact that neither the timespan, nor the average air temperature between lateral flow events contribute significantly to either of the two models has interesting implications for the question as to where the P in our lateral flow samples originally came from. Precipitation is unlikely to be the main source, as typical concentrations of $P_{tot}$ in throughfall are much lower than in our lateral flow samples (Sohrt et al. 2017). Also, since higher flow rates are not associated with higher $P_{tot}$ concentrations, transport of particulate matter in the organic layer does not appear to be a critical process for $P_{tot}$ transport. What remains as a possible source for the $P_{tot}$ in our lateral flow samples, is leaching of highly 280 transportable P forms from organic layer material. This leaching is thought to occur through degradation of organic layer material by plants and microbes which convert P bound to organic matter to soluble forms. Since microbial activity is heavily influenced by water availability and temperature, we expected periods with higher temperatures to be associated with higher $P_{tot}$ concentrations in lateral flow. Our results do not show such a relation with mean air temperature, though, higher soil moisture does lead to higher P concentrations in lateral flow. The fact that the time 285 spans between events are not significantly correlated to $P_{tot}$ concentrations in lateral flow indicates, that little or no accumulation of water transportable P in the organic layer occurs in between flow events.

## Conclusions

The generation of lateral flow in the organic layer in response to rainfall is very heterogeneous among our study sites, 290 and so is the $P_{tot}$ transport associated to it. Average $P_{tot}$ concentration in lateral flow was not proportional to P concentration or -content of the organic layer material at the respective site. This strongly suggests, that generalizations about lateral flow and the associated P transport from case studies into a broader context have to be treated with caution. This is empathized by our model results in which the site ID, which may also serve as a site specific error term, was by far the most significant parameter regarding P transport in the organic layer.

The fact that the set of environmental data we used to explain the P transport only accounted for 20-30 % explained variability demonstrates, that our understanding of its underlying causes is still limited. What our results clearly show is, that simple scenarios for P transport in this context are not feasible. Neither water flow as the mobilizing force, nor accumulation of transportable P in the organic layer in between events can be supported as primary regulators for P transport in lateral flow by our study. Our advice to future researchers on this topic would be to focus on detailed case 300 studies and controlled experiments in the field, rather than on large spatial and temporal coverage. This would allow for the use of artificial tracers as well as the distinction between different P species in lateral flow, which may help in determining the drivers of organic layer P transport with later flow.






Code availability

The authors do not plan to make the code underlying this study publicly available.

Data availability

Upon acceptance of the manuscript for publication, the complete raw data the study is based on will be deposited in a suitable public data repository.

Sample availability

The samples used in this study are not IGSN-registered.


Appendices and supplements

There are no appendices and supplements for this publication.

Author contributions

Jakob Sohrt collected the data used in this study with the exception of meteorological data, which was  provided by the federal forest research institutes of Bavaria and Baden-Württemberg. Jakob Sohrt takes responsibility for the integrity of the data and the accuracy of the data analysis. All listed authors comply with the journals authorship policy.

Conflict of interest

The authors declare no conflict of interest.

Acknowledgments

This study was carried out under a grant from the DFG funded project SPP 1685 Ecosystem Nutrition, Forest Strategies for limited Phosphorus Resources under the project ID WE 4598/7-1. On site information on meteorological conditions
and soil moisture was kindly provided by the federal forest research institutes of Bavaria and Baden-Württemberg.  We thank Delon Wagner, Ruth Magh, Franziska Zieger and Lisa Dankwerth for their help with field work and sample handling.




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





Table 1 Stand parameters of our field sites.

| site | Vessertal (VES) | Mitterfels (MIT) | Conventwald (CON) |
|---|---|---|---|
| location | 50°36'20.8"N 10°46'17.1"E | 48°58'33.4"N 12°52'33.5"E | 48°01'15.9"N 7°57'56.1"E |
| mean precipitation (mm a$^{-1}$)[a] | 1200 | 1229 | 1749 |
| mean temperature (°C)[a] | 5.5 | 4.7 | 6.6 |
| elevation (m asl)[a] | 810 | 1023 | 884 |
| vegetation[b] | Fagus sylvatica 94 %, Picea abies 2 %, Abies alba 2 % | Fagus sylvatica 100 % | Fagus sylvatica 69 %, Abies alba 31 % |
| stand age (a)[a] | 120 | 110 | 100-150 |
| slope angle at trench site (°) | 15 | 8 | 22 |
| slope exposition | east-south-east | west | south |
| humus form[b] | moder | moder | more-like moder |
| texture topsoil[b] | loam | loam | loam |
| soil stone content %[b] | 63 | 25 | 69 |
| geology[a] | Trachyandesite | Paragneiss | Paragneiss |
| org. layer thickness (cm)[b] | 14 | 8 | 13 |
| org. layer mass (t ha$^{-1}$)[b] | 139.3 | 50.5 | 117.4 |
| org. layer $P_{tot}$ conc. (µg g$^{-1}$)[b]* | 1333.7 | 1380.7 | 1147.7 |
| mineral soil $P_{tot}$ conc. (µg g$^{-1}$)[b]* | 991.7 | 906.8 | 581.9 |
| org. layer P pool (kg ha$^{-1}$)[b] | 193.39 | 69.03 | 159.53 |
| total soil P pool to 1 m (kg ha$^{-1}$)[b] | 464 | 678 | 231 |

a: (Haußmann and Lux 1997), original source not available, cited from (Zavišić et al. 2016)

b: (Lang et al. 2017), *: mass weighted average calculated from supplementary material from (Lang et al. 2017)



*Table 2 Length of observation period, total observed precipitation and lateral flow at the three study sites during the period.*

| Site | | MIT | VES | CON |
|---|---|---|---|---|
| Period | | 2015-04-01 to 2015-10-19 | 2015-05-08 to 2015-10-10 | 2015-02-23 to 2015-11-21 |
| Period length | days | 201 | 155 | 271 |
| Total precipitation | mm | 523 | 421.4 | 549.4 |
| Total lateral flow | l | 1409 | 496.8 | 205.6 |




Table 3 Parameterization *of multiple linear models to predict the $P_{tot}$ concentration in lateral flow. The full parameter set as well as the optimized parameter selection according to minimized AIC and their corresponding estimate, p-value and individual explained variability are shown. Parameters written in bold are included in the final model structure.*

*Total model $R^2$ was 0.58 and 0.70 for the intra-event and inter-event model respectively, total model adj. $R^2$ was 0.57 and 0.67 respectively.*

| Intra-event data model see fig. 3 A | estimate | p-value | expl. var. % | Inter event data model see fig. 3 B | estimate | p-value | expl. var. % |
|---|---|---|---|---|---|---|---|
| **site MIT (intercept)** | 2.03005 | < 2e-16 | | **site MIT (intercept)** | -0.312676 | 0.554926 | |
| **site VES** | -0.2832 | 9.08e-8 | 35.95 | **site VES** | 0.355078 | 0.052979 | 41.41 |
| **site CON** | 0.59732 | < 2e-16 | | **site CON** | 0.95509 | 4.35e-08 | |
| **flow rate log10 (l s⁻¹)** | 0.07284 | 0.0398 | 1.53 | mean flow rate log10 (l s⁻¹) | | | |
| | | | | **max flow rate log10 (l s⁻¹)** | -0.20103 | 0.012913 | 8.88 |
| **cum. event volume log10 (l)** | -0.2124 | < 2.1e-11 | 8.69 | event flow sum log10 (l) | | | |
| **EC log10 (µSi cm⁻¹)** | 0.18218 | 7.67e-11 | 4.1 | **mean EC log10 (µSi cm⁻¹)** | 0.981421 | 1.45e-05 | 18.59 |
| soil moisture (%) | | | | mean soil moisture (%) | | | |
| mean air temp. since last event (°C) | | | | mean air temp. since last event (°C) | | | |
| **mean soil moisture since last event (%)** | 0.04006 | < 2e-16 | 7.78 | **mean soil moisture since last event (%)** | 0.037857 | 0.000115 | 4.79 |
| time since last event (h) | | | | time since last event (h) | | | |
| **Complete model** | | < 2.2e-16 | 58.07 | **Complete model** | | 1.001e-11 | 70.55 |



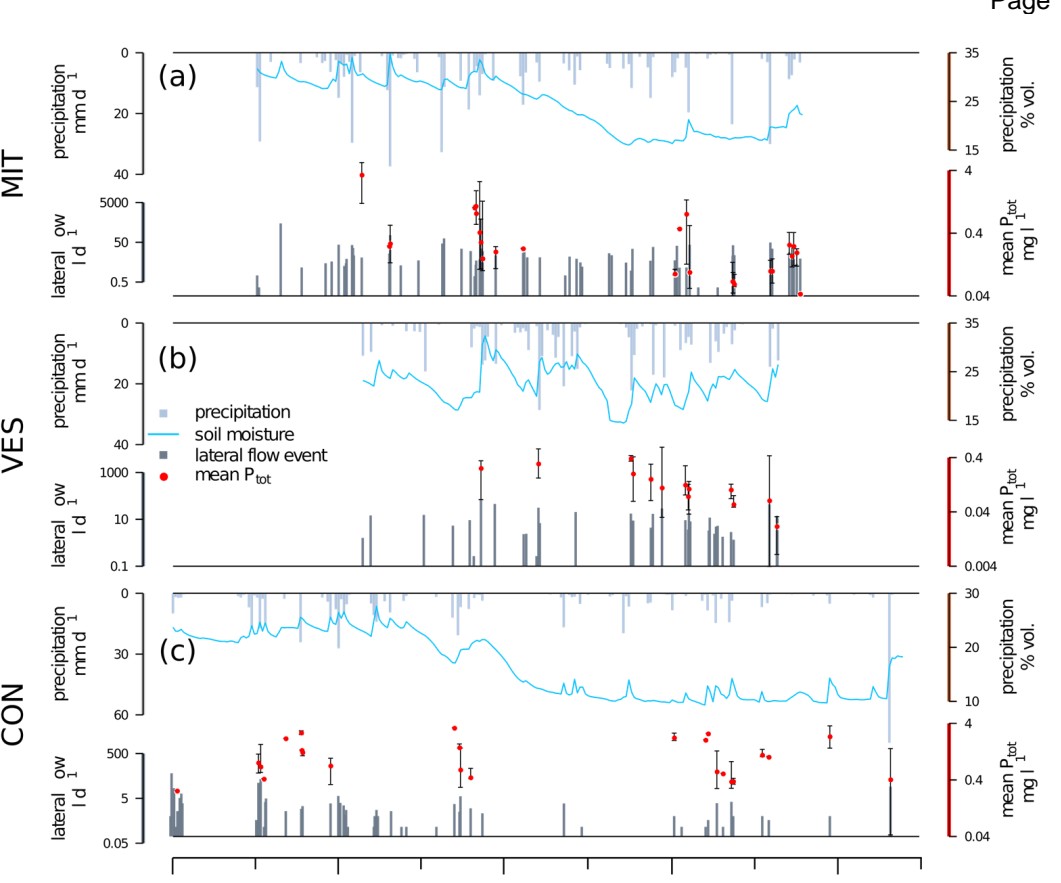

*Figure 1* Overview of precipitation, soil moisture, rate of lateral flow, mean (red dots) and range (error bars) of $P_{tot}$ concentration in lateral flow at the study sites Mitterfels (a), Vessertal (b) and Conventwald (c) during the observation period.





*Figure 2* *Hydrological conditions and P transport during exemplary lateral flow events at the sites CON (a), MIT (b, c) and VES (d, e). Individual sample measurement uncertainty for $P_{tot}$ is given as black bar, cumulative uncertainty for $P_{tot}$ load as orange area behind the respective symbols.*






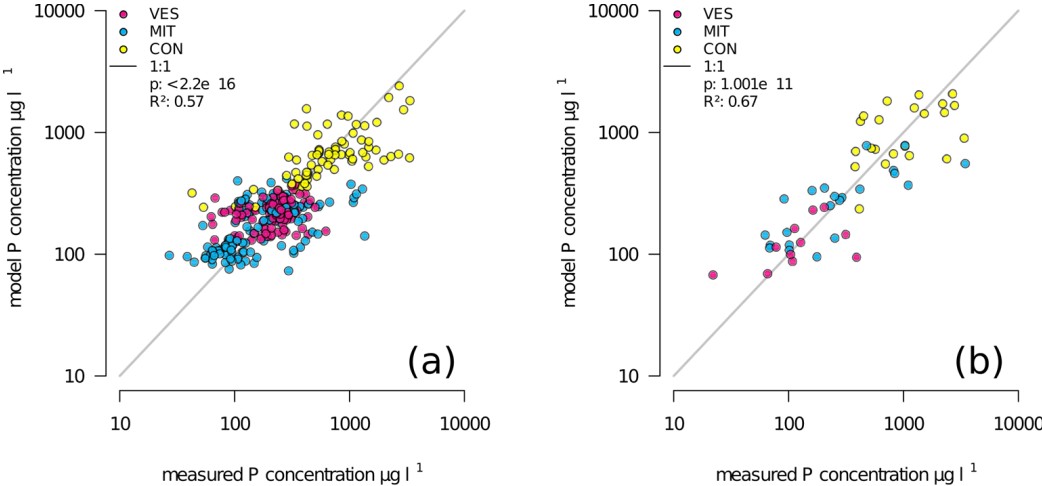

Figure 3 *Intra-event data (a) and inter-event data data (b) of the $P_{tot}$ concentration in lateral flow, and results of the best-fit multiple linear model constructed from our data set. For model parameterization, p-values and explained variability per parameter see Tab. 3.*