# Peer review of "Phosphorus transport in lateral subsurface flow at forested hillslopes"

_SOIL, 2018_

## Referee Comment (RC1) · Anonymous Referee #1 · 5 Jul 2018

Review of manuscript soil-2018-13 entitled "**Phosphorus transport in lateral subsurface flow at forested hillslopes by Sohrt et al.**"

This study involves the automated temporal collection of lateral flow (flux) of inter and intra rainfall events from three sites over 6-9 months which was used to estimate total Phosphorus and EC of collected water. Authors also measured soil moisture separately from these 3 sites. Later on authors used this information to develop multiple linear models for total P dynamics in relation to intra- and inter-flow events. The topic is interesting and focuses on the phosphorus transport along the hillslopes in the forested soils. Authors have introduced the topic very well and included the related recent works to show the need of the present research. However, the methodology is not clearly defined and results and discussion are poorly described. There is no mention what are the modelling parameters, how Authors obtained these and how model respond to the soil, precipitation, vegetation variables at different sites. In fact the study lacks in generating appropriate soil, climate and vegetation parameters to draw any suitable relation between the soluble phosphorus dynamics along the forested hillslope. I do not see any novel contribution in the paper apart from automated collection of temporal lateral flow which is already published elsewhere.

1. How authors selected the location for the flow monitoring at three sites, what criteria adopted and how their measurements are related to hillslope or soil texture or rainfall amount/intensity/duration or vegetation at different sites?
2. Authors must clarify how they segregated overland flow from the subsurface lateral flow? Probably, subsurface lateral flow generation and P concentration also depends on the soil texture, clay content, water holding capacity, soil heterogeneity as well as the extent of organic matter in the soils. Authors are invited to explain how their measures are related to these soil characteristics?
3. Authors didn't describe how soil moisture was estimated (L148-150), how often and why it was measured at 2m upslope (L116) only?
4. Authors mention (L118) the collection of samples at 2 and 3 weeks intervals for total P measurements but it is not clarified how they estimated intra and inter-event P concentration from these measurements?
5. It is not clear how authors constructed the linear models, must elaborate and include the linear mathematical equations. How authors interpolated P concentration for each intra and inter-events? How P contents are related to flow volume or EC or air temperature, seems doubtful. Additionally, the estimation of flow events based on subsurface lateral flow volume is doubtful. If there was no relation between different measures (L197-198), how authors developed linear relationship between P concentration and other measures (L152-153). How a precipitation event of 1mm (L174) can generate measurable flow which is generally ignored while estimating effective rainfall?
6. There was opportunity to measure other soluble nutrients, why authors restricted themselves to measure only the total P concentration while they could have done much more from the collected flow samples.

**Other comments**

L148    what was the relevance of measuring air temperature?

L173    not leafs but leaves

L177 among (at?)?

L185-86 how these event were separated from the follow generation events?

L 213 what was the most measureable precipitation event?

L217 coincide with

L224-29 Did authors measured hydrophobicity of organic matter? Seems irrelevant justification. Rather authors should have estimated the soil texture, soil heterogeneity, vertical and horizontal hydraulic conductivity, amount of organic matter etc. which could have more valid relation with the measured data.

L251 I'm unable to see R2 and Adj R2 in Table 2, how authors estimated the adjusted R2 values and why?

L255 what is AIC, authors must explain their model and input parameters.

L259 where is the data to substantiate this conclusion. I do not see any such relation in the Figs. 2, 3 or 4. I'm unable to see the model parameters. What authors mean by significant association here, did they applied any statistical test on their data?

Table 1 If organic layer thickness was only 8cm at MIT then why the measurements were made between 10-25 cm soil depth?

Table 3 there is lot of confusion in the models here and Fig 3. How authors estimated model P concentration? I'm unable to see what inputs required for both models, how they obtained? How authors estimated the data shown in Table 3. Why log values for flow rate and other parameters shown here?

Fig 1 I do not see the soil moisture axis in any of the Figs, what you mean by precipitation vol and lateral ow? Why soil moisture decreases as the precipitation increases over the time in (a) and (c)?

Fig. 2 what authors mean by exemplary flow here? If P estimation was made on biweekly or triweekly basis (L118-121) then how authors obtained intra-event P concentration.

---

## Author Comment (AC1) · 11 Jul 2018

* * *
**Authors' response:**

Dear reviewer, thank you for taking the time to review the manuscript. We feel that there have been some general misunderstandings, which we hope will be cleared up by our proposed changes to the manuscript based on your comments.

In the following response to the review, the reviewers' comments are written in regular black text, the authors reply to a respective comment starts with a "->" and is written in blue, the proposed change in the revised manuscript starts again with a "->" is written in green, if necessary a quote of the proposed text in the revised manuscript is posted in black italics.

Line numbers in the answers to the reviewers' comments generally relate to the original submission file.
* * *
**Review:**

This study involves the automated temporal collection of lateral flow (flux) of inter and intra rainfall events from three sites over 6-9 months which was used to estimate total Phosphorus and EC of collected water.

➔ I am not sure if there has been some kind of general misunderstanding here: We did not sample the rainfall (although the rainfall intensity was measured), we directly sampled the lateral flow in the org layer and topsoil. We captured the lateral flow in a drainage system (L 101-105), continually measured its flow rate and EC (L 105-109) and drew flow proportional samples in a temporal resolution of 10-60 min (L 109-112), which were then individually analyzed in the lab for P (L 125-147).

Authors also measured soil moisture separately from these 3 sites.

➔ Our trench installation for capturing lateral flow is 10 m wide; soil moisture measurements were made 2 m upslope of the trench. 2 m is hardly a meaningful separation in this context. The 2 m distance was kept intentionally to avoid disturbing the soil material at the point of sample collection by creating permanent flow paths along the moisture sensors.

Later on authors used this information to develop multiple linear models for total P dynamics in relation to intra-and inter-flow events. The topic is interesting and focuses on the phosphorus transport along the hillslopes in the forested soils. Authors have introduced the topic very well and included the related recent works to show the need of the present research. However, the methodology is not clearly defined and results and discussion are poorly described.

➔ We recognize that there is a need here to clarify some points, which we detail further in the answers to your individual comments.

➔ The section in "methods" regarding model parameterization and fitting will be completely reworked: *"For each hillslope site, time series of lateral flow rate, $P_{tot}$ concentration in lateral flow, electric conductivity (EC), air temperature and soil moisture in the organic layer are compiled for the observation period. In addition to the measured parameters, the dataset contains some calculated indices: Average soil moisture and air temperature in between the last and present flow event as well as the time duration of that period. This dataset, which contains the $P_{tot}$ measurements in their full resolution and corresponding values of all other parameters, is called the "full resolution dataset" from here on. To identify environmental variables that may*

*explain the variability in P concentration in the lateral subsurface flow and to test, whether similar relations can be established for all three sampling sites, a multiple linear model with the $P_{tot}$ concentration in lateral flow as the dependent variable is utilized. The IDs of the three experimental sites are included as a categorical variable, which can also serve as an error term. Those parameters that are not normally distributed ($P_{tot}$ conc., flow rate, EC and cum. event volume) are transformed accordingly before the model fitting. From this full set of parameters, the parameter selection for the model is carried out by stepwise forward selection/backwards elimination of variables with respect to the Akaike Information Criterion (AIC, see Akaike (1973)) through the step()-function in the R base-environment (R-Core-Team 2014). The relative parameter importance was calculated according to (Grömping 2006). The model fitted to the "full resolution dataset" is consequently called the "full resolution model".*

*To test whether our high resolution sampling approach is worthwhile, the same analysis is performed with an aggregated version of the otherwise same dataset: In order to simulate a bulk sampling approach, the full resolution dataset is aggregated contain only one value of each parameter per flow event.. Depending on the parameter this entails sums (flow volume), arithmetic averages (flow rate, soil moisture, air temperature) or flow weighted averages (EC, $P_{tot}$) of the original data. Flow events are defined according to two criteria; If only a single peak in lateral flow rate occurs within a distinct precipitation event and until the cessation of lateral flow, the whole period is regarded as a single event. If multiple flow peaks occur within a period of uninterrupted flow, the period may be split into multiple events. If the flow rate falls below 10 % of the preceding flow peak before increasing again and if this increase corresponds to a prior increase in rainfall intensity, this is used as a cutoff point between events. The resulting dataset is called "bulk event dataset" from here on, the resulting multiple linear model "bulk event model"*

For better understanding, the terms "intra-event dataset/model" and "inter-event dataset/model" will be changed to "full resolution dataset/model" and "bulk event dataset/model" throughout the manuscript

There is no mention what are the modeling parameters,

➔ The model parameters are (among other places within the manuscript) listed in Tab 3

how Authors obtained these

➔ This is discussed in detail in the methods section

and how model respond to the soil, precipitation, vegetation variables at different sites. In fact the study lacks in generating appropriate soil, climate and vegetation parameters to draw any suitable relation between the soluble phosphorus dynamics along the forested hillslope.

➔ We intentionally focus the study on those parameters were we have sufficient data to actually test the effects of those certain parameters on lateral flow P concentrations. The chance of finding statistically significant relations between lateral flow P concentrations and specific site characteristics are very slim, as that would mean that we only have 3 data points to compare. However, we do recognize that that site specific effects exist, which is why we included the site ID as a categorical variable into the multiple liner model, which can be understood as a site specific error term (see L 293-294) that would represent the sum of all site characteristics that affect P concentrations in lateral flow.

I do not see any novel contribution in the paper apart from automated collection of temporal lateral flow which is already published elsewhere.

➔ We kindly ask the reviewer to provide a source to a peer-reviewed study that has sampled and analyze P in lateral flow at a temperate forest sites over multiple months in a sub-hourly temporal resolution. To the best of our knowledge, no such (or even similar) study exists thus far, making this manuscript a novel contribution to the scientific community.

**Individual Comments:**

1.

How authors selected the location for the flow monitoring at three sites, what criteria adopted and how their measurements are related to hillslope or soil texture or rainfall amount/intensity/duration or vegetation at different sites?

➔ The approximate location of the sites was a consequence of them being the defined research sites of the project this study was part of (see L 79-82). The approximate site selection was carried out in order to find sites of similar vegetation with differences in soil P content (see L 80-82). Soil texture and rainfall characteristics where not used as site selection criteria.
The exact spot was determined by practical reasons, namely the questions "what places are accessible with an excavator?" and "can we get permission to dig there?".

2.

Authors must clarify how they segregated overland flow from the subsurface lateral flow?
Probably, subsurface lateral flow generation and P concentration also depends on the soil texture, clay content, water holding capacity, soil heterogeneity as well as the extent of organic matter in the soils. Authors are invited to explain how their measures are related to these soil characteristics?

➔ We did not segregate overland flow from subsurface flow, the revised manuscript will include that clarification. We did not concentrate on the differences in soil parameters between the sites as a possible explanation for differences in P transport or lateral flow generation for one simple reason: Three sites (= three data points) is far too little to draw any meaningful conclusion in my opinion. It was for that reason that we instead focused on the effect of parameters that are not constant and therefore offer more data points. However, we do recognize that that site specific effects exist, which is why we included the site ID as a categorical variable into the multiple liner model, which can be understood as a site specific error term (see L 293-294) that would represent the sum of all site characteristics that affect P concentrations in lateral flow.

➔ In the method section we will explicitly state that there was no distinction between overland flow and subsurface flow in the organic layer: "*In the present method there is no distinction between overland flow and subsurface flow in the organic layer.*"
Throughout the manuscript, "lateral subsurface flow" has will be rephrased to "lateral flow" if appropriate.

3.

Authors didn't describe how soil moisture was estimated (L148-150), how often and why it was measured at 2m upslope

➔ Soil moisture was measured with TOMST TMS-3 loggers in 5 min intervals. It was measured 2 m upslope in order to not disturb the collection of lateral flow. The revised manuscript will include that clarification

➔ The revised manuscript will contain a description of soil moisture measurements. "*Soil moisture in the organic layer is measured with TOMST TMS-3 probes in 3 locations, 2 m up slope of each trench installation. The measurements take place in a depth of 20 cm in a temporal resolution of 5 min.*"

(L116)only?

➔ The word "only" is being used intentionally. It is supposed to mean, that besides the parameter total P, no other form of P is analyzed, so that "only" total P is analyzed.

4.

Authors mention (L118) the collection of samples at 2 and 3 weeks intervals for total P measurements but it is not clarified how they estimated intra and inter-event P concentration from these measurements?

➔ Samples were drawn into the autosamplers at ~10-30 min intervals depending on the flow rate and stored within the autosamplers, each of which can store up to 40 samples (L 110-111).The autosamplers were emptied bi- or triweekly, each then containing up to 40 distinct samples. L118-121 will receive clarification so that this cannot be confused anymore. Regarding the distinction of single events see also L 154-157.

➔ The revised manuscript will include a clarification regarding this point. "*The autosamplers were emptied in the field twice a week at the site CON and every three weeks at the sites VES and MIT, then containing up to 40 samples each.*"

5.

It is not clear how authors constructed the linear models, must elaborate and include the linear mathematical equations.

➔ The procedure for construction the model is straightforward and laid out in L 163-170. The Equation for the models can be derived from table 3 which includes all required values.

How authors interpolated P concentration for each intra and inter-events?

➔ P concentrations for the intra-event model are not interpolated in any way but are rather the measured raw data. P concentrations for the inter-event model are flow-weighted averages of the measured P concentrations within a single event (L 153). The definition of "event" in this context can be found in L 154-157

How P contents are related to flow volume or EC or air temperature, seems doubtful.

➔ We kindly ask the reviewer to clarify what exactly "seems doubtful". Also we do not claim a significant relation of lateral flow P concentrations to air temperature (see table 3).

Additionally, the estimation of flow events based on subsurface lateral flow volume is doubtful.

➔ We kindly ask the reviewer to clarify this comment.

If there was no relation between different measures (L197-198), how authors developed linear relationship between P concentration and other measures (L152-153).

➔ L197-198 was meant to be a visual description of the data plotted in fig. 2. Key here are the words "*not immediately recognizable*" and "*consistent*". This will be clarified further in the revised manuscript. The calculated multiple linear relationship between the model parameters and P concentrations does not require a relationship that is either consistent or visually obvious.

➔ The revised manuscript will be amended in order to clarify that the text in questions is supposed to be a visual description of fig. 2. "*The variation of $P_{tot}$ concentrations in the full resolution dataset displayed in fig 2 appears more ordered, with recognizable periods of increase or decrease.*"

How a precipitation event of 1mm (L174) can generate measurable flow which is generally ignored while estimating effective rainfall?

➔ A false positive in our lateral flow measurements seem highly unlikely to me. However, we do recognize that our placement of precipitation gauges was unfortunate in at least on the three sites, which we discussed in L 231-240. A possible explanation for some cases at least would thus be that the precipitation at the lateral flow measurement site was higher than at the precipitation measurement site.

6.

There was opportunity to measure other soluble nutrients, why authors restricted themselves to measure only the total P concentration while they could have done much more from the collected flow samples.

➔ I had the equipment, budget, lab time and staff to measure P, which was also the aim of the project I was employed in. Every research projects exists within some sort of constraints.

Other comments:

L148 what was the relevance of measuring air temperature?

➔ The fundamental nature of the experiment (as a general scientific concept) dictates that the significance of single parameters for the result may reveal itself only after the experiment has been concluded. In this case, air temperature turned out to be not a relevant parameter for our modeling approach (see Tab. 3).

L173 not leafs but leaves

➔ Will be changed accordingly

L177 among (at?)?

→ As far as I know "among" (in the sense of "in between)" is the correct term here.

L185-86 how these event were separated from the follow generation events?

→ See L154-157

→

L 213 what was the most measureable precipitation event?

→ If you mean by that the largest amount of precipitation within one day then that was ~75 mm at the site CON in late Nov.

L217coincide with

→ Will be changed accordingly

L224-29

Did authors measured hydrophobicity of organic matter? Seems irrelevant justification. Rather authors should have estimated the soil texture, soil heterogeneity, vertical and horizontal hydraulic conductivity, amount of organic matter etc. which could have more valid relation with the measured data.

→ We did not measure hydrophobicity, nor soil texture or hydraulic conductivity, so that our statements on that matter remain speculative, which is implied by the fact that the statement in questions is written as subjunctive. For the reasons cited in our answer to comment 2, we did not prioritize measuring site specific conditions and rather focused on parameters that exhibit some temporal variability.

L251 I'm unable to see R2 and Adj R2 in Table 2, how authors estimated the adjusted R2 values and why?

→ Thank you for pointing this out, the reference here was supposed to be table 3, this will be changed in the revised manuscript. Adj. $R^2$ was calculated with the lm() function, which is part of the R-base package. The benefit of comparing $R^2$ and adj. $R^2$ is that a large difference between the two can be a hint that a model is over-parameterized.

→ Will be changed accordingly. "*The high significance of both models and the minor difference between $R^2$ and adj. $R^2$ in the respective models (see Tab. 3) make it appear unlikely, that the additional explained variability in the bulk event data model is simply due to a reduction in sample size, or over-parametrization.*"

L255

what is AIC, authors must explain their model and input parameters.

→ The AIC is the "Akaike Information Criterion"(see L 163). Thank you for pointing out the missing citation/source, it will be added in the revised manuscript.

➔ The revised manuscript will contain the necessary citation. "*From this full set of parameters, the parameter selection for the model is carried out by stepwise forward selection/backwards elimination of variables with respect to the Akaike Information Criterion (AIC, see Akaike (1973)) through the step()-function in the R base-environment (R-Core-Team 2014).*"

L259
where is the data to substantiate this conclusion. I do not see any such relation in the Figs. 2, 3 or 4. I'm unable to see the model parameters. What authors mean by significant association here, did they applied any statistical test on their data?
➔ The data to support this claim can be found in table 3, in the line labeled "mean soil moisture since last event (%)", which is supposed to be understood to mean the same as "antecedent soil moisture". The term "significant" here is to be understood as "associated with a p-value < 0.05", which is given in the same line of table 3. Since this apparently was unclear, a reference to table 3 will be given at the appropriate place.
➔ A reference to Tab. 3 will be included at the respective place.

Table 1
If organic layer thickness was only 8cm at MIT then why the measurements were made between 10-25 cm soil depth?
➔ 8 cm is an average thickness over a large area determined by another study at this site (see table 1); the actual thickness of the org. layer may vary over distances much shorter than the 10 m length of the trenches. Due to practical constraints it was not always possible to exactly match the drainage to the local thickness of the org layer.
➔ I also think there has been a misunderstanding, I do not say that measurements were made in between 10 and 25 cm, I just say that this is the variation in thickness of the org layer found at the three trench sites. The lower end of the drained portion of soil was matched to the lower end of the organic layer as well as possible und field conditions.

Table 3
there is lot of confusion in the models here and Fig 3. How authors estimated model P concentration? I'm unable to see what inputs required for both models, how they obtained?
➔ Regarding the estimation of model P concentrations: The multiple linear model was first constructed of all the parameters shown in table 3. According to the AIC the model structure was optimized (see L 162-170) in order to reduce the amount of parameters, since a lot of them are cross-corelated. The parameters in table 3 printed in bold are those that remained in the final model structure after model optimization (aimed at minimizing the AIC). The estimated model P concentration is what the model then produces for a certain point in time (of the observation period) with the model parameters how they were measured at that point in time.
➔ The left part of table 3 contains the model parameter names, their estimate, p-value and relative importance (here as a partial R², see L 170) explained by the model for the intra event model, the

right half of the table contains the same information for the inter-event model. All of this is already written out in the table caption and the table heading. Do you see the need to amend it?

→ The section in "methods" regarding model parameterization and fitting will be completely reworked (see quote in the section "review"). For better understanding, the terms "intra-event dataset/model" and "inter-event dataset/model" will be changed to "full resolution dataset/model" and "bulk event dataset/model" throughout the manuscript

How authors estimated the data shown in Table 3.

→ I do not really understand the question I think. Table 3 shows what it says in the caption: "Parametrization of multiple linear models…". It contains the names of the parameters, their estimate and p-value (which you regularly get when fitting a linear model) and their relative importance (here given as a partial $R^2$) as detailed in the methods section (L 163-170).

→ The section in "methods" regarding model parameterization and fitting will be completely reworked (see quote in the section "review"). For better understanding, the terms "intra-event dataset/model" and "inter-event dataset/model" will be changed to "full resolution dataset/model" and "bulk event dataset/model" throughout the manuscript

Why log values for flow rate and other parameters shown here?

→ If the data behind a parameter is not normally distributed, it should be transformed into a normal distribution before inclusion into the multiple linear model. Otherwise some single values of non-normally distributed parameters may impose a great effect on the overall model. Those parameters that were originally not normally distributed (but rather log-distributed in this case) were transformed accordingly.

→ Some text will be added explaining the reasoning behind transforming some of the time series before fitting the model. "*Those parameters that are not normally distributed ($P_{tot}$ conc., flow rate, EC and cum. event volume) are transformed accordingly before the model fitting*"

Fig 1

I do not see the soil moisture axis in any of the Figs, what you mean by precipitation vol and lateral ow?
Why soil moisture decreases as the precipitation increases over the time in (a) and (c)?

→ There has indeed been some mix up in the labeling of this plot. It so supposed to say "lateral flow". The soil moisture axis was mislabeled as "precipitation" and is supposed to be in the top right y-axis. That is also why the unit here is written as "% vol". This will be changed in the revised manuscript. With regards to your second question I must say that I do not see what you describe in the subplots a and c. Why do you think that precipitation is increasing? The summer of 2015 was in fact extremely dry at these two sides compared to the long term average.

→ The updated fig. 1 will be added to the revised manuscript

Fig. 2

what authors mean by exemplary flow here? If P estimation was made on biweekly or triweekly basis (L118-121) then how authors obtained intra-event P concentration.

→ Samples were drawn into the autosamplers at ~10-60 min intervals, depending on the flow rate and stored within the autosamplers, the autosamplers were emptied bi- or triweekly, then containing up to 40 samples. The respective section will receive clarification so that this cannot

be confused anymore. What you see in fig. 2 is the data from these ~10-60 min sampling resolution.

➔ The revised manuscripts will contain a clarification regarding this point. "*The autosamplers were emptied in the field twice a week at the site CON and every three weeks at the sites VES and MIT, then containing up to 40 samples each.*"

---

## Referee Comment (RC2) · Anonymous Referee #2 · 20 Aug 2018

This manuscript aims to improve the understanding of processes responsible for P transport in lateral flow in forest soils. As the authors correctly points out there are very few data of the kind, and for this reason this manuscript is of interest to readers of Soil. It is also well written. Having said that, it is a pity that only total-P was determined in the sampled waters, and not e.g. PO4-P, DOC and other chemical parameters, which might have added more insight of what P that is mobilized when. Naturally this limits the extent of the increased understanding that is possible with this experimental setup. Apart from that, my most important remarks are the following:

- From the data provided, it is not possible to understand what kind of soil this represents. Please supply more basic soil data, most importantly the soil classification, organic C and soil pH. - Lines 100-110 (approx.). To construct the sampling device,

a trench was dug in the soil. Such a disturbance could potentially cause nitrification and other undesirable side effects initially. Please explain if and how such effects were considered, and whether e.g. the first sample was collected after a certain time period to avoid possible disturbance effects - Just as reviewer I, I was also wondering what water the authors actually collected (i.e. overland or subsurface flow), and the reply from the authors does not really provide any clues. Please consider this once more, this may be significant for the overall interpretation of the results.

Other minor remarks

- Line 70 – "transport capacity transport". Sounds strange - Style: Mix of past and present tense at places, try to be consistent - Fig. 1. This graph with multiple entries is a bit hard to understand at first, it would be easier if it was broken up in two separate figures, with a smaller number of dependent variables - Fig. 3 caption. "data data". One "data" too many? - Line 217, "that do not coincident", should be "that do not coincide"

---

## Author Comment (AC2) · 6 Sep 2018

* * *
**Authors' response:**

Dear reviewer (2), thank you for taking the time to review and comment upon the manuscript. We carefully considered your input and revised the respective parts of the manuscript accordingly. We hope that this will help to improve the quality of the manuscript and address your constructive criticism.

In the following response to the review, the reviewers' comments are written in regular black text, the authors reply to a respective comment starts with a "->" and is written in regular blue, the proposed change in the revised manuscript starts again with a "->" is written in green, if necessary a quote of the proposed text in the revised manuscript is posted in black italics.

Line numbers in the answers to the reviewers' comments generally relate to the original submission file.
* * *
**Remarks:**

From the data provided, it is not possible to understand what kind of soil this represents. Please supply more basic soil data, most importantly the soil classification, organic C and soil pH. - Lines 100-110 (approx.).

➔ More comprehensive soil information will be added to table 1, which in the revised version will also contain the following lines:

| site | Vessertal (VES) | Mitterfels (MIT) | Conventwald (CON) |
|---|---|---|---|
| Soil classification (WRB 2015)[b] | Hyperdystric skeletic chromic cambisol (Hyperhumic. Loamic) | Hyperdystric folic cambisol (Arenic. Loamic. Nechic. Protospodic) | Hyperdystric skeletic folic cambisol (Hyperhumic. Loamic) |
| Soil pH (H$_2$O) 0-5 cm[b] | 2.8 | 3 | 3.2 |
| Soil C 0-5 cm (mg g$^{-1}$)[b] | 18 | 18 | 22 |

*b: Lang et al. 2017*

To construct the sampling device, a trench was dug in the soil. Such a disturbance could potentially cause nitrification and other undesirable side effects initially. Please explain if and how such effects were considered, and whether e.g. the first sample was collected after a certain time period to avoid possible disturbance effects.

➔ At lines 84-85 there is written "Sample collection was carried out from March to November 2015 with the construction of the field installations taking place the year before."

➔ This will be changed to include the following:

*"Sample collection was carried out from March to November 2015 with the construction of the field installations taking place the year before or earlier (CON: Sept-Nov 2013, MIT: Mar-May 2014, VES: Jun-Sept 2014). It is assumed that the period in between construction of the trenches and the beginning of the sampling campaign was sufficient to allow for equilibration of the relevant biochemical processes."*

Just as reviewer I, I was also wondering what water the authors actually collected (i.e. overland or subsurface flow), and the reply from the authors does not really provide any clues. Please consider this once more, this may be significant for the overall interpretation of the results.

➔ The revised version of the manuscript based on the suggestions by reviewer 1 contains the following sentence in the methods-section: "In the present method there is no distinction between overland flow and subsurface flow in the organic layer."

➔ This will be changed to include the following:

*"In the present method there is no distinction between overland flow and subsurface flow in the organic layer. This is due to the fact that the organic layer is made up of fragmented materials without a contiguous surface are which prevents the possibility of discrimination between surface runoff and subsurface flow in this soil layer. For this reason, this type of flow is sometimes referred to as biomat flow (Sindle et al. 2007)."*

**Minor remarks:**

Line 70 – "transport capacity transport".

➔ This was a mistake on our part. The third word will be deleted in the revised manuscript.

Style: Mix of past and present tense at places, try to be consistent

➔ Past tense will not be used in the revised manuscript.

Fig. 1. This graph with multiple entries is a bit hard to understand at first, it would be easier if it was broken up in two separate figures, with a smaller number of dependent variables

➔ We considered the possibility of breaking this graph into two graphs but decided against it for the following reason: It is important so see all variables together, since the aim is for the reader to see that there is no immediate connection between P concentrations and the other environmental variables on a seasonal scale.

➔ However, we to take your input serious and will take measures to improve the readability of the graph. This includes added spacing in between the three subplots and more accessible labeling of the experimental sites with label the three subplots:

Fig. 3 caption. "data data". One "data" too many?

➔ Yes, this will be changed in the revised manuscript

Line 217, "that do not coincident", should be "that do not coincide"

➔ This will be changed accordingly in the revised manuscript